# CRISPR screens are feasible in *TP53* wild-type cells

Kevin R Brown[1] iD, Barbara Mair[1] iD, Martin Soste[1] iD & Jason Moffat[1,2,3] iD

Comment on: **E Haapaniemi *et al*** (July 2018)
See reply: **E Haapaniemi *et al*** (August 2019)

A recent study in *Nature Medicine* (Haapaniemi *et al*, 2018) caused concern amongst researchers and investors alike after reporting that genome editing with CRISPR may be less efficient in human cells with intact p53 signalling.

The conclusion by Haapaniemi *et al* that the p53-mediated DNA damage response (DDR) hampers CRISPR-based functional genomic dropout screens inspired us to investigate this phenomenon more closely. When performing a genome-scale CRISPR screen in RPE1 cells, harbouring wild-type *TP53*, Haapaniemi *et al* reported both a failure to detect essential genes and the enrichment of single-guide RNAs (sgRNAs) targeting *TP53*, *CDKN1A* (encoding p21) and *RB1* (pRb). This enrichment of sgRNAs targeting the p53 axis was not observed in a p53-deficient RPE1 cell line, where essential genes were depleted as expected. From this, together with a number of follow-up experiments, the authors conclude that the p53 pathway is activated by CRISPR-induced double-strand breaks (DSBs) leading to a cell cycle arrest. Therefore, this presents an obstacle to precision genome editing as well as functional genomic screening in *TP53* wild-type cells. This hypothesis would be consistent with previous research showing that a single DSB is sufficient for inducing a prolonged p53-dependent arrest in normal human fibroblasts (Di Leonardo *et al*, 1994; Linke *et al*, 1996). It is also in line with the complementary findings by Ihry *et al* (2018) reporting that the p53-dependent response to Cas9 toxicity reduces the efficiency of human pluripotent stem cell (hPSC) genome engineering and indicating that TP53 status should be monitored in hPSC before and especially after engineering.

We and others have performed genome-wide screens in *TP53* wild-type and p53-deficient cell lines, including RPE1 cells (Hart *et al*, 2015, 2017; Zimmermann *et al*, 2018). In agreement with Haapaniemi and colleagues, sgRNAs targeting *TP53*, *CDKN1A*, *RB1* and other p53 pathway genes became enriched over the course of the RPE1 screen, while sgRNAs targeting the negative p53 regulators *MDM2* and *MDM4* were depleted (Fig 1A). Similar observations were also made in other *TP53* wild-type cell lines, including HCT116 colon cancer cells and a glioblastoma cell line, whereas HeLa cells (non-functional p53 due to expression of the E6 viral oncogene) and DLD1 cells (oncogenic $TP53^{S241F/-}$ mutation Ahmed *et al*, 2013; Sur *et al*, 2009) do not show this effect. While the assertion by Haapaniemi *et al* is that DSBs induced by Cas9 editing drive p53 activation in the wild-type background, this is not directly measured in their (or our) experiments. In fact, similar enrichments have been observed in large-scale RNAi experiments, which do not rely on the induction of DSBs (Giacomelli *et al*, 2018).

Haapaniemi *et al* also report that sgRNAs targeting a set of ribosomal genes, largely expected to be essential for cell growth, failed to deplete in the wild-type *TP53* RPE1 background, whereas they dropped out as expected in their p53-deficient RPE1 cells. From this, the authors conclude that a transient cell cycle arrest mediated by p53 explains the failure to detect essential genes in the CRISPR screen. In contrast, we find that gene-level dropout profiles are highly correlated between all pairs of cell lines ($P < 2.2 \times 10^{-16}$), regardless of their p53 status (Fig 1B), arguing against p53 hampering the depletion of essential genes.

We and others have developed methods to evaluate the performance of functional genomics screens (Hart *et al*, 2014, 2015, 2017), which have been used to quality-control different sgRNA libraries (Sanson *et al*, 2018). Applying these to a published RPE1 screen (Hart *et al*, 2015), we find that sgRNAs targeting essential genes such as the ribosome or proteasome subunits, or "gold-standard essential genes" (Hart *et al*, 2017), deplete as expected (Fig 1C) and similar to cell lines with non-functional *TP53*. These results demonstrate that screens in immortalized and cancer cell lines with wild-type *TP53*, including RPE1, still identify essential genes targeted by the depleted sgRNAs.

To ensure that our results were not unique to a small or biased dataset, or to a specific CRISPR library, we examined the dropout profiles of essential gene sets across a compendium of 517 genome-wide CRISPR screens in various cancer cell lines (Meyers *et al*, 2017) (DepMap18Q4 version; https://depmap.org/portal/) or 14 genome-wide screens in AML cell lines (Wang *et al*, 2017). We found negligible differences in the fold-change distributions of sgRNAs targeting these gene sets between *TP53* wild-type and different categories of *TP53* mutant cell lines in the DepMap screens (Fig 1D, data not shown for AML cell lines). Precision–recall curves based on the gold-standard gene sets show very high performance (i.e. high precision and recall and therefore a large area under the curve, AUC) across all DepMap screens, with a mean AUC of 0.99, regardless of p53 status. Critically, the DepMap screens were performed using the

---

1  Donnelly Centre, University of Toronto, Toronto, ON, Canada. E-mail: j.moffat@utoronto.ca
2  Department of Molecular Genetics, University of Toronto, Toronto, ON, Canada
3  Canadian Institute for Advanced Research, Toronto, ON, Canada
**DOI** 10.15252/msb.20188679 | Mol Syst Biol. (2019) 15: e8679

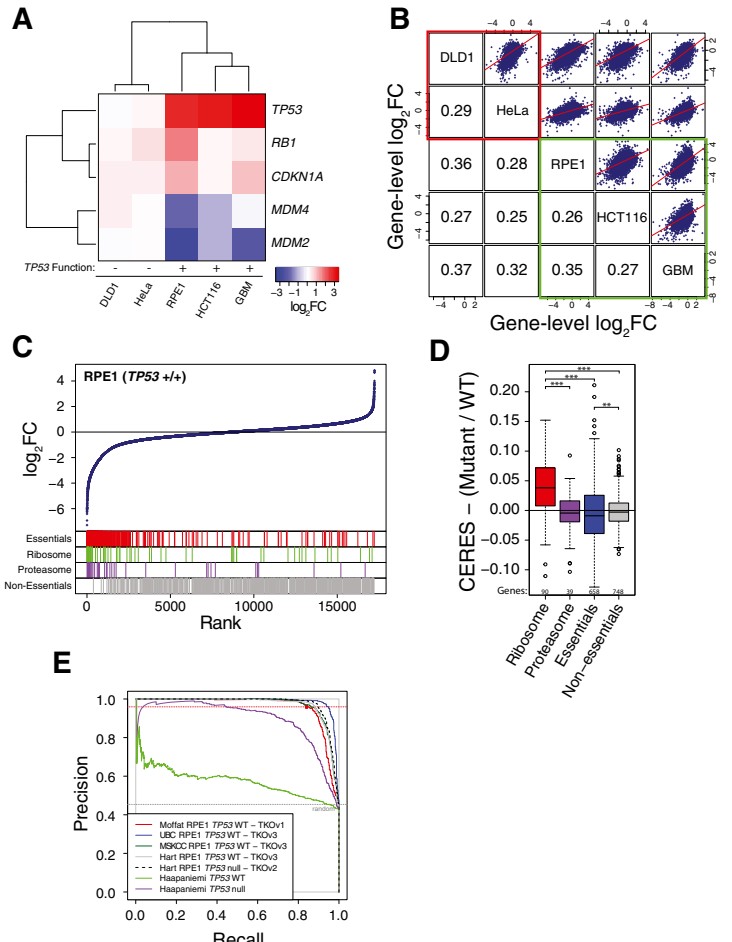

**Figure 1. Performance of CRISPR–Cas9 functional genomic screens with respect to *TP53* status.**
(A) Genome-wide CRISPR–Cas9 screens were performed in five established human cell lines. Heatmap shows mean log$_2$-fold change of all sgRNAs targeting the indicated p53 pathway genes. (B) Mean log$_2$-fold change scatterplots for all 17,236 genes targeted by the TKOv1 library for screens described in (A). Bottom triangle indicates Spearman correlation between screen pairs. Red: cell lines with non-functional p53. Green: cell lines with functional p53. (C) Mean log$_2$-fold changes of all sgRNAs per gene are shown for the results of a CRISPR screen with *TP53* wild-type RPE1 cells. Gene sets comprising gold-standard essential genes (essentials), ribosome or proteasome subunits, or gold-standard non-essential genes (non-essentials) are highlighted. (D) Ratios of CERES scores for reference gene sets between *TP53* wild-type and mutant/non-functional cell lines across 356 cancer cell lines in the 18Q4 Avana (DepMap) dataset for which *TP53* functional status was available in Giacomelli *et al* (***FDR < 0.001, **FDR < 0.01, Wilcoxon rank sum test with multiple testing correction). (E) Precision–recall curves based on the mean log$_2$-fold change of sgRNAs using gold-standard essential and non-essential genes across five RPE1 *TP53* wild-type genome-wide CRISPR screens and two RPE1 *TP53*-deficient ("null") genome-wide CRISPR screens. The CRISPR libraries and source of the screens are indicated. Also see main text for details and references.

the same analyses of four additional published or unpublished RPE1 genome-wide loss-of-function CRISPR screens performed in three different laboratories, including one screen carried out in a *TP53*-deficient background (Zimmermann *et al*, 2018) (Fig 1E). Our original RPE1 screen (red) and the other *TP53* wild-type and *TP53*-deficient RPE1 screens show high performance, indicating that, regardless of *TP53* status, recovery of the expected essential genes is feasible in RPE1 cells. In contrast, the screen in *TP53*-deficient RPE1 cells from Haapaniemi *et al* shows moderate performance, while the screen in *TP53* wild-type RPE1 cells failed to identify gold-standard essential genes with any precision. The authors argue that this is due to p53 signalling; however, that conclusion is inconsistent with the RPE1 *TP53* wild-type screens from other groups. In our experience, such underperformance is more likely due to poor editing efficiency. Haapaniemi *et al* selected their cells using functional editing of the *HPRT1* gene, which does not require high or sustained Cas9 activity, and they did not directly report editing efficiency of their cell line after the initial selection process.

Haapaniemi *et al* and Ihry *et al* have reported an important phenomenon in the context of CRISPR–Cas9 genome editing, and we echo support for monitoring *TP53* status, especially with respect to cell engineering for clinical applications. However, we suggest key steps that should not be ignored when publishing any CRISPR screen results:

1. Prior to performing a CRISPR screen, editing efficiency in the cell line system to be screened should be assessed using accepted protocols such as the SURVEYOR assay or editing tests with essential genes. Low editing efficiency can easily yield results for positive selection screens, but is not compatible with measuring sgRNA dropout in pooled screens.

2. Comprehensive assessment of screen performance should be conducted using established benchmarks and metrics, such as reference standard essential gene sets for pooled dropout screens.

3. Observations should be validated in independent cell lines or clones, preferably with orthogonal methods or through genetic "rescue" experiments to establish the robustness of the reported findings.

Avana gRNA library (Doench *et al*, 2016), indicating that our findings are not specific to our own CRISPR libraries. Similarly, Ihry *et al* (2019) reported successful genome-scale CRISPR screens in a *TP53* wild-type human pluripotent stem cell line, despite observing considerable p53-mediated Cas9 toxicity in these cells (Ihry *et al*, 2018), which is in agreement with data from our own laboratory in a comparable system

(Mair *et al*, 2019). In summary, while the induction of a p53 response in *TP53* wild-type cells is not unexpected, it does not hamper the screenability of *TP53* wild-type cells in general.

Finally, we obtained the raw data from Haapaniemi *et al*, allowing us to generate precision–recall curves using the gold-standard essential gene set (Hart *et al*, 2017) to assess screen performance. We incorporated

The application of gene editing technologies to genome-wide functional genomics provides an incredible opportunity to accelerate functional annotation of the human genome with high precision and accuracy. While we do not disagree that CRISPR may evoke a p53 response in certain genotypes, examination of hundreds of published genome-wide CRISPR screens conducted independently by multiple laboratories in different geographic locations showed that major concerns are not warranted when performing CRISPR screens in wild-type *TP53* cells. Therefore, we would like to reinforce the notion of existing functional genomics standards to quality-control genome-scale screening data in order to avoid some of the pitfalls that were discovered during the early years of RNA interference screens.

## Data availability

Complete code and data files are provided as Dataset EV1.

RPE1 screening data: Gene Expression Omnibus accession number GSE128210 (https://www.ncbi.nlm.nih.gov/geo/query/acc.cgi?acc=GSE128210).

**Expanded View** for this article is available online.

## Author contributions

KRB and BM initiated the study and wrote the manuscript, KRB performed the data analysis, MS provided critical insight and interpretation to the work and assisted with editing the manuscript, and JM supervised the work, provided critical insight and helped edit the manuscript.

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
