## [Review Process File · Molecular Systems Biology]

CRISPR screens are feasible in *TP53* wild-type cells

Kevin R. Brown, Barbara Mair, Martin Soste, Jason Moffat.

Review timeline:

Submission date:	8 th October 2018
Editorial Decision:	15 th November 2018
Revision received:	18 th December 2018
Accepted:	1 st February 2019

Editor: Maria Polychronidou

Transaction Report:

1st Editorial Decision

15th November 2018

Thank you again for submitting your work to Molecular Systems Biology. We have now heard back from the three referees who agreed to evaluate your manuscript. As you will see below, the reviewers think that the presented findings are relevant for the CRISPR field and they are overall supportive of publication. They raise however some concerns, which we would ask you to address in a revision.

Overall, I think that the reviewers' recommendations are clear and there is therefore no need to repeat the points listed below. Please let me know in case you would like to discuss in further detail any of the issues raised.

REFeree REPORTS

Reviewer #1:

In their correspondence, Brown et al. address the effect of the TP53 mutation status in RPE1 cells on CRISPR/Cas9 editing efficacy in response to a publication by Haapaniemi et al. in Nature Medicine (2018). The manuscript by Brown et al. specifically report on their meta-analysis of pooled CRISPR-Cas9 based negative selection screens in response to conclusions raised in Haapaniemi et al. (and to a minor extent in Ihry et al., 2018). Haapaniemi et al. described that double stranded breaks introduced by CRISPR-Cas9 can trigger a p53 mediated apoptotic response of TP53 positive cells which limits genome-editing events. While the authors agree that a p53 mediated cellular response can have an effect on genome-editing based applications, the authors disagree with the conclusions of the previous study that screens in TP53 positive cell lines using CRISPR/Cas9 are severely hampered. To this end, the authors summarise the results of the previous studies and then describe their approach for the meta-analysis of the original study's raw data accompanied by data from their own lab and also data from large screening efforts by independent laboratories.

From this analysis, Brown et al conclude that the observation by Haapaniemi and colleagues can be explained by sub-optimal quality of the screens rather than the TP53 mutation status of the cells. This is an important contribution and a note of caution on the interpretation of screening data.

Specific points:

- The authors should clarify which metric (fold changes, Bayes Factors?) the ROC curve displayed in Figure 1E is based upon. For the sake of transparency it is important that all screening data and computer code to reproduce Figure 1 are published with the manuscript.
- The addition of a screen TP53 WT that has not been carried out using a TKO library in figure 1E) would increase its generality
- "completely amenable" is scientifically not defined and should be rephrased
- Something went wrong with the last sentence of the abstract

Reviewer #2:

General comments:

Overall, this manuscript makes a very important statement in the field of CRISPR-Cas9. It is widely accepted that a single Cas9-generated DSB is sufficient to trigger a p53-dependent DNA damage response in TP53 WT cells, reinforcing the importance of monitoring TP53 status for CRISPR-based therapeutic applications. However, contrasting to what has been reported in Haapaniemi et al, TP53 WT cells are still amenable to genetic screens, provided that the necessary functional standards are met. The importance of always running these standard tests for all published screens is highlighted in the manuscript and is a very relevant point.

Specific comments:

1. In this manuscript, the authors show that sgRNAs targeting TP53 (and associated genes) are highly enriched in viability screens, across different TP53-WT cell lines (Fig 1A). This corroborates the p53-induced cell cycle arrest mediated by Cas9-breaks. However, the TP53 status does not alter the overall distribution of sgRNAs (Fig. 1B) and the authors show that the depletion of essential genes is still comparable between TP53 WT and mutant cell lines (Fig 1C).

In addition to Figure 1C, I would want to see the same density plot depicting sgRNAs targeting all essential genes (Published in Hart et al 2017 G3), instead of the ones targeting only the proteasome and ribosome genes.

2. Next, taking a more systematic approach, the authors examined the dropout profiles of essential genes (using the Hart dataset), across a compendium of 436 genome-wide screens in various cancer cell lines. Here, they used the DeepMap dataset, instead of the Project Achilles. Project Achilles is a massive effort from the Broad Institute to catalogue essentiality profiles of different cell lines using CRISPR-screens performed with a wide range of libraries. I would want to see how the data would look using the Achilles dataset.

3. Finally, in Fig 1E, they generate ROC curves to measure the performance of several screens on distinguishing between essential and non-essential genes, concluding that the screen performed by Haapaniemi et al had intrinsic flaws associated with incomplete editing. Here, I think it's important to:

A. Clarify which dataset from Haapaniemi et al is being analyzed. By reading the methods of this paper, it's clear that the authors performed a focused screen (around 12 000 sgRNAs for which they show no data and a genome-wide screen using the Brunello library. In the manuscript, they mention a selection of cells using functional editing of the HPRT1 gene, which was used for the focused screen. However, as far as I am aware the dataset available from Haapaniemi et al is from the genome-wide screen. It would be important to clarify this.

B. Comment on how the use of different libraries can contribute for this discrepancy of results. Is it because the sgRNAs for the Brunello library do not produce the editing efficiency that is necessary to see essential genes dropping out in TP53-WT cells? It doesn't seem to be the case based on Fig 1D, but it would be important to know the authors opinion on this, specially if this manuscript is to transmit a message to the CRISPR community.

Overall, this is a very interesting and relevant paper. It would be ideal if the authors tried to reproduce exactly what has been done previously by other groups ie show data from a CRISPR-screen performed in RPE TP53-WT and mutant cells, with the Brunello library, and exactly the same setup as the one done in Haapaniemi et al. However, I understand this is beyond the scope of current manuscript.

Reviewer #3:

Two recent papers in Nature Medicine report that a single cut induced by Cas9 is sufficient to activate DNA damage response mediated by p53. Ihry et al. show that this causes toxicity in human embryonic stem cells, while Haapaniemi et al. report transient cell cycle arrest in immortalized RPE-1 cells. Furthermore, the latter paper claims that this arrest hampers prediction of genes whose loss leads to cell death or decreased cell proliferation. The paper under the evaluation here, specifically takes issue with this last claim from Haapaniemi et al. The authors first evaluate overall fold-change values for sgRNA in p53 WT and p53 mutant cancer cell lines and claim that there are no noticeable differences between them. They proceed to analyze differences in essential gene prediction among 436 p53 WT and mutant cancer cell lines and conclude that, again, there are no major differences between the two groups. Finally, the authors perform precision-recall analysis of core essential genes performed in p53 WT and null RPE-1 screens, showing the subpar performance of Haapaniemi et al. screens compared to screens performed on the same cells by other groups. This leads the authors to conclude that CRISPR/Cas9 screens are feasible in p53 WT cells.

In the view of this reviewer, the manuscript raises reasonable doubt about conclusions reached by Haapaniemi et al. and rightfully asks for more rigorous quality control testing of CRISPR screens. However, while an important finding of the study (Figure 1E) is convincing, other conclusions need to be drawn from more carefully performed analysis. Only then the paper could be a timely contribution to the current controversy surrounding the application of CRISPR/Cas9 technology in p53 WT cells. The major concerns in detail are:

- 1.) The authors confound p53 point-mutants and p53 null cells throughout the manuscript. While p53 mutants are generally less capable of activating p21, there is a range of activating capacities among them (and this capacity for the same mutant differs in different cell lines); thus, mutant p53 could potentially exhibit some (albeit weaker) of the same effects described in p53 WT cells. This is especially important in figure 1D, where the proper comparison should be between p53 WT and p53 null cell lines.
- 2.) The inclusion of MDM2/MDM4 data in figure 1A is misleading. Since MDM2/MDM4 depletion leads to cell cycle arrest/apoptosis in the absence of DNA damage as well, it cannot be taken as proof of Cas9 activation (unlike the enrichment of TP53 or CDKN1A that can only be ascribed to activation due to DNA damage). Thus, it should be removed from the figure.
- 3.) Figure 1B is too general to be conclusive. Comparing overall log₂FC in this way shows very little. What kind of change would be significant in the authors' opinion? This data would be better represented as a scatter plot matrix of all cell lines against all, with correlation coefficients for log₂FC values between each pair. This way one could easily see if these correlations are higher for cell lines within the group of the same p53 status or between the two groups.
- 4.) While I find figure 1E convincing, I cannot help but think its logic is in certain way circular. The claim of Haapaniemi et al. is that there will be a differential prediction of essential genes in p53 WT and null cells when CRISPR/Cas9 screens are performed, and that accurate prediction will be possible only in p53 null cells. However, the set of core essential genes in this manuscript is largely based on the results from p53 WT and cells that express different mutant forms of p53, with only one cell line being null for p53 (HL-60). Thus, it is important that the authors show that there is no difference in predicting essential genes in p53 WT and null cells, which should be done in figure 1D.
- 5.) The wording of several sentences needs to be more careful. In the last sentence of paragraph 7, as well as in the second sentence of the last paragraph, the authors claim that screenability of TP53 WT cells is not hampered by CRISPR/Cas9 technology. This claim ignores findings from Ihry et al. All the data collected here is based on cancer cell lines and immortalized lines. The situation is very different in hESCs, and potentially in primary cells. Thus, I would suggest amending those instances of "TP53 wild-type cells" to "TP53 wild-type cancer and immortalized cell lines". This concerns the manuscript title as well.

Of minor note: HeLa cells are described in the manuscript as "mutant", though they have WT p53 (albeit nonfunctional due to other reasons). This should be corrected.

Reviewer #1:

In their correspondence, Brown et al. address the effect of the TP53 mutation status in RPE1 cells on CRISPR/Cas9 editing efficacy in response to a publication by Haapaniemi et al. in Nature Medicine (2018). The manuscript by Brown et al. specifically report on their meta-analysis of pooled CRISPR-Cas9 based negative selection screens in response to conclusions raised in Haapaniemi et al. (and to a minor extent in Ihry et al., 2018). Haapaniemi et al. described that double stranded breaks introduced by CRISPR-Cas9 can trigger a p53 mediated apoptotic response of TP53 positive cells which limits genome-editing events. While the authors agree that a p53 mediated cellular response can have an effect on genome-editing based applications, the authors disagree with the conclusions of the previous study that screens in TP53 positive cell lines using CRISPR/Cas9 are severely hampered. To this end, the authors summarise the results of the previous studies and then describe their approach for the meta-analysis of the original study's raw data accompanied by data from their own lab and also data from large screening efforts by independent laboratories.

From this analysis, Brown et al conclude that the observation by Haapaniemi and colleagues can be explained by sub-optimal quality of the screens rather than the TP53 mutation status of the cells. This is an important contribution and a note of caution on the interpretation of screening data.

Specific points:

- The authors should clarify which metric (fold changes, Bayes Factors?) the ROC curve displayed in Figure 1E is upon. For the sake of transparency it is important that all screening data and computer code to reproduce Figure 1 are published with the manuscript.

We used gene level log₂ fold-change (LFC) values (i.e. mean of all sgRNA fold-change values for each gene) for all of our analyses in Figure 1, except for Figure 1D where CERES scores (Meyers et al, Nature Genetics 2017, PMID 29083409) were used to calculate relative differences across the dropout behaviour of gene sets in cell lines with either normal or different categories of mutant TP53. We have now clarified this in the figure legend. The code to reproduce all the Figure panels will be available online (e.g. journal link and lab webpage) and also by request.

- The addition of a screen TP53 WT that has not been carried out using a TKO library in figure 1E) would increase its generality

We agree with the reviewer that it is critical to evaluate screens performed with additional libraries to ensure that findings are generalizable and robust. This was included in the original submission by examining trends in the DepMap data from the Broad Institute, and also the 14 AML screens performed by Wang et al. The DepMap screens were carried out across 517 cancer cell lines using the Avana library (in the 4Q18 release), while the Wang paper used their own genome-wide library, and through these analyses we demonstrated that our findings were not specific to the TKO library.

We have now incorporated a deeper analysis of the DepMap data by integrating the p53 status of each of the cell lines as determined in Giacomelli et al (Nature Genetics 2018, PMID 30224644) as follows. Across the range of DepMap screens where p53 status is available (N = 427), the mean UC using the same PR-curve approach is 0.99, regardless of p53 status (WT N = 105; WT non-functional N = 31; mutant functional = 40; mutant non-functional N = 251). This compares to an AUC of 0.9 for Haapaniemi's p53-deficient RPE1 screen, and 0.57 for the WT p53 RPE1 screen. Since the DepMap screens are carried out using the Avana library, the PR analysis is not biased by the CRISPR library being employed, and the large number of screens of both WT p53 and mutant/non-functional p53 backgrounds further supports our original claim that p53 status does not significantly impair detection of fitness genes by CRISPR. We have now modified the second paragraph on page 2 to reflect this additional analysis.

- "completely amenable" is scientifically not defined and should be rephrased

- Something went wrong with the last sentence of the abstract

We thank the reviewer for pointing this out and have rephrased/fixed these issues this in the "standfirst text" that replaces the abstract.

Reviewer #2:

General comments:

Overall, this manuscript makes a very important statement in the field of CRISPR-Cas9. It is widely accepted that a single Cas9-generated DSB is sufficient to trigger a p53-dependent DNA damage response in TP53 WT cells, reinforcing the importance of monitoring TP53 status for CRISPR-based therapeutic applications. However, contrasting to what has been reported in Haapaniemi et al, TP53 WT cells are still amenable to genetic screens, provided that the necessary functional standards are met. The importance of always running these standard tests for all published screens is highlighted in the manuscript and is a very relevant point.

Specific comments:

1. In this manuscript, the authors show that sgRNAs targeting TP53 (and associated genes) are highly enriched in viability screens, across different TP53-WT cell lines (Fig 1A). This corroborates the p53-induced cell cycle arrest mediated by Cas9-breaks. However, the TP53 status does not alter the overall distribution of sgRNAs (Fig. 1B) and the authors show that the depletion of essential genes is still comparable between TP53 WT and mutant cell lines (Fig 1C). In addition to Figure 1C, I would want to see the same density plot depicting sgRNAs targeting all essential genes (Published in Hart et al 2017 G3), instead of the ones targeting only the proteasome and ribosome genes.

We would like to draw the reviewer's attention to the top stripchart in Figure 1C, which shows the gold standard essential gene set from Hart et al. The ribosome and proteasome genes are a subset of these, and were shown mainly because they were used as reference sets in the Haapaniemi manuscript.

2. Next, taking a more systematic approach, the authors examined the dropout profiles of essential genes (using the Hart dataset), across a compendium of 436 genome-wide screens in various cancer cell lines. Here, they used the DeepMap dataset, instead of the Project Achilles. Project Achilles is a massive effort from the Broad Institute to catalogue essentiality profiles of different cell lines using CRISPR-screens performed with a wide range of libraries. I would want to see how the data would look using the Achilles dataset.

Our understanding is that the CRISPR screens in Project Achilles and the Broad portion of the DepMap Project are in fact the same data. The Achilles data portal will be completely transferred to the DepMap portal starting Winter 2018 (<https://portals.broadinstitute.org/achilles>). Our analyses use the most up-to-date dataset in the Achilles/DepMap portal.

3. Finally, in Fig 1E, they generate ROC curves to measure the performance of several screens on distinguishing between essential and non-essential genes, concluding that the screen performed by Haapaniemi et al had intrinsic flaws associated with incomplete editing. Here, I think it's important to:

A. Clarify which dataset from Haapaniemi et al is being analyzed. By reading the methods of this paper, it's clear that the authors performed a focused screen (around 12 000 sgRNAs for which they show no data and a genome-wide screen using the Brunello library. In the manuscript, they mention a selection of cells using functional editing of the HPRT1 gene, which was used for the focused screen. However, as far as I am aware the dataset available from Haapaniemi et al is from the genome-wide screen. It would be important to clarify this.

We agree with the reviewer that there was confusion about what data and what screens were being discussed. The Methods section in the Haapaniemi paper is unclear with respect to this, and screen data are not publicly available. Therefore, we directly contacted the authors, who kindly provided the primary data from their genome-wide screen. They clarified that they used the Brunello sgRNA library in HPRT1-edited RPE1 TP53 wild-type cells (non-clonal, derived from a "fairly large selected pool") and in a clonal TP53-deficient RPE1 cell line with a defined mutation in TP53 (Sokolova et al, Cell Cycle 2017, PMID 27929715), but to our understanding not HPRT1-selected. The "Cas9 HPRT-version of it is not clonal" (correspondence with Jussi Taipale, 25.07.2018).

B. Comment on how the use of different libraries can contribute for this discrepancy of results. Is it because the sgRNAs for the Brunello library do not produce the editing efficiency that is necessary to see essential genes dropping out in TP53-WT cells? It doesn't seem to be the case based on Fig 1D, but it would be important to know the authors opinion on this, specially if this manuscript is to transmit a message to the CRISPR community.

We agree with the reviewer that in principle, different libraries can account for discrepant results.

However, we do not believe that the Brunello library is the underlying cause for the screens to fail in the Haapaniemi paper, and given that there were no controls shown to support the editing efficiency in the model, we can only speculate as to other reasons for this failure. In fact, a manuscript from the Doench lab on bioRxiv (<https://www.biorxiv.org/content/early/2018/07/02/356626>) shows that in properly controlled dropout screens, the Brunello and TKOv3 libraries have very comparable performances based on AUC values. The analyses we have presented in this submission are consistent across three different libraries, hundreds of cell lines and multiple labs, and clearly contradict the findings from the Haapaniemi N = 1 screen.

Overall, this is a very interesting and relevant paper. It would be ideal if the authors tried to reproduce exactly what has been done previously by other groups ie show data from a CRISPRscreen performed in RPE TP53-WT and mutant cells, with the Brunello library, and exactly the same setup as the one done in Haapaniemi et al. However, I understand this is beyond the scope of current manuscript.

We agree with the reviewer that this is an interesting and relevant experiment, but also agree that it is outside the scope of the manuscript. In lieu of this, we provided robust and consistent results across a large compendium of published data with multiple libraries. Furthermore, we would like to reiterate that we included screens in both TP53 WT (parental cell line) and p53-deficient RPE1 cells, so the only variable we did not assess was the Brunello library, which, as mentioned above, we would expect to have very similar performance as the TKOv3 library.

Reviewer #3:

Two recent papers in Nature Medicine report that a single cut induced by Cas9 is sufficient to activate DNA damage response mediated by p53. Ihry et al. show that this causes toxicity in human embryonic stem cells, while Haapaniemi et al. report transient cell cycle arrest in immortalized RPE-1 cells. Furthermore, the latter paper claims that this arrest hampers prediction of genes whose loss leads to cell death or decreased cell proliferation. The paper under the evaluation here, specifically takes issue with this last claim from Haapaniemi et al. The authors first evaluate overall fold-change values for sgRNA in p53 WT and p53 mutant cancer cell lines and claim that there are no noticeable differences between them. They proceed to analyze differences in essential gene prediction among 436 p53 WT and mutant cancer cell lines and conclude that, again, there are no major differences between the two groups. Finally, the authors perform precision-recall analysis of core essential genes performed in p53 WT and null RPE-1 screens, showing the subpar performance of Haapaniemi et al. screens compared to screens performed on the same cells by other groups. This leads the authors to conclude that CRISPR/Cas9 screens are feasible in p53 WT cells.

In the view of this reviewer, the manuscript raises reasonable doubt about conclusions reached by Haapaniemi et al. and rightfully asks for more rigorous quality control testing of CRISPR screens. However, while an important finding of the study (Figure 1E) is convincing, other conclusions need to be drawn from more carefully performed analysis. Only then the paper could be a timely contribution to the current controversy surrounding the application of CRISPR/Cas9 technology in p53 WT cells. The major concerns in detail are:

1.) The authors confound p53 point-mutants and p53 null cells throughout the manuscript. While p53 mutants are generally less capable of activating p21, there is a range of activating capacities among them (and this capacity for the same mutant differs in different cell lines); thus, mutant p53 could potentially exhibit some (albeit weaker) of the same effects described in p53 WT cells. This is especially important in figure 1D, where the proper comparison should be between p53 WT and p53 null cell lines.

We thank the reviewer for their critical reading of our submission, and agree that precise description of the p53 status is important to correctly interpret the work. We have carefully inspected and revised the use of the terms “wild-type”, “null” and “mutant” throughout the manuscript. For Figure 1D, we have made use of an analysis published in Giacomelli et al (Nature Genetics 2018; PMID 30224644), where the genetic and functional status of p53 was evaluated for more than 1000 CCL cell lines, including 427 that were profiled by CRISPR in the DepMap dataset. We have integrated their classifications into our analyses of the DepMap data in order to assess CRISPR function in cells that are WT and functional for p53 signaling, as well as cell lines that are mutant and non-functional. As detailed in our response to reviewer 1, the performance of CRISPR dropout screens is not significantly different (FDR > 0.1) between these two classes, with a mean AUC of 0.99 for both

WT/functional and mutant/non-functional (assessed using precision-recall curves and the gold standard essentials gene set from Hart et al, G3 2017, PMID 28655737). We have revised Figure 1D to specifically compare these two classes, rather than simply “WT” and “mutant”.

2.) The inclusion of MDM2/MDM4 data in figure 1A is misleading. Since MDM2/MDM4 depletion leads to cell cycle arrest/apoptosis in the absence of DNA damage as well, it cannot be taken as proof of Cas9 activation (unlike the enrichment of TP53 or CDKN1A that can only be ascribed to activation due to DNA damage). Thus, it should be removed from the figure.

We appreciated this comment, so we significantly reworded the first section of the paper for clarification. Even though Haapaniemi *et al.* conclude from the observed enrichment of sgRNAs targeting TP53, CDKN1A (or others as noted by Giacomelli et al, Nature Genetics 2018, PMID 30224644) during the screen that DSBs caused by CRISPR induce a p53 response, our point with Figure 1A is just to show that also in our screens disruption of p53 pathway components in a wild-type background leads to enrichment of the corresponding sgRNAs (or depleted for negative regulators like MDM2 and MDM4). These effects are completely lost in cells that have lost functional TP53. But as CRISPR screens do not measure protein abundance or activation, neither we nor Haapaniemi *et al.* can state definitively that p53 is being activated by CRISPR-induced DSBs just from a CRISPR screen result. While it is likely that this is indeed the case, the remainder of our manuscript shows that regardless of p53 pathway activity, CRISPR screens work perfectly well in both WT and mutant/non-functional p53 backgrounds. Therefore, we have left Figure 1A as it was originally presented, but clarified both the main text and the figure legend describing it.

3.) Figure 1B is too general to be conclusive. Comparing overall log2FC in this way shows very little. What kind of change would be significant in the authors' opinion? This data would be better represented as a scatter plot matrix of all cell lines against all, with correlation coefficients for log2FC values between each pair. This way one could easily see if these correlations are higher for cell lines within the group of the same p53 status or between the two groups.

We thank the reviewer for their feedback, and have now changed Figure 1B to the suggested gene-level scatter plot matrix complete with Spearman correlation values. We observe a statistically significant correlation between each pair of these screens, even between cell lines with different p53 status. Furthermore, while we agree that Figure 1B on its own may not be conclusive, it supports our position that WT TP53 does not inhibit the feasibility of CRISPR screens. We think that in combination with our analyses of more than 400 screens of the latest DepMap release as well as the 14 AML screens in the Wang paper, these observations present compelling and comprehensive support for our conclusions.

4.) While I find figure 1E convincing, I cannot help but think its logic is in certain way circular. The claim of Haapaniemi et al. is that there will be a differential prediction of essential genes in p53 WT and null cells when CRISPR/Cas9 screens are performed, and that accurate prediction will be possible only in p53 null cells. However, the set of core essential genes in this manuscript is largely based on the results from p53 WT and cells that express different mutant forms of p53, with only one cell line being null for p53 (HL-60). Thus, it is important that the authors show that there is no difference in predicting essential genes in p53 WT and null cells, which should be done in figure 1D.

We have performed a comprehensive precision-recall analysis across the Broad's DepMap screens, described in the third paragraph on page 2, focusing solely on cell lines that were shown to be functional for TP53 signaling, or mutant and non-functional (based on the analysis in Giacomelli et al, Nature Genetics 2018, PMID 30224644). Using the same gene sets that failed to drop out in the Haapaniemi screen (ribosomal genes, proteasomal genes), or our gold-standard set, we replotted Figure 1D to show there is little difference between the p53 classes (largely similar to the original Figure 1D). Below you will also find a boxplot of all the AUC values from our PR analysis across all four p53 classes, showing uniformly high AUC values regardless of p53 status. Hence, there is no difference ($FDR > 0.1$) in measuring the dropout of essential genes between cell lines of different TP53 status.

While the Haapaniemi paper suggests that their CRISPR screens are hampered in TP53 WT cells, the correct experiment to prove this would have been to ‘rescue’ TP53 in their TP53-null cells and show that these cells cannot be screened.

5.) The wording of several sentences needs to be more careful. In the last sentence of paragraph 7, as well as in the second sentence of the last paragraph, the authors claim that screenability of TP53 WT cells is not hampered by CRISPR/Cas9 technology. This claim ignores findings from Ihry et al. All the data collected here is based on cancer cell lines and immortalized lines. The situation is very different in hESCs, and potentially in primary cells. Thus, I would suggest amending those instances of "TP53 wild-type cells" to "TP53 wild-type cancer and immortalized cell lines". This concerns the manuscript title as well.

We agree with the reviewer that careful and accurate wording is essential in this matter. We would like to mention that we do reference the Ihry *et al.* manuscript and agree with their conclusions. Importantly however, the Ihry *et al.* paper did not indicate a complete lack of screenability, and the same group posted a separate paper in bioRxiv describing genome-wide screens in human ES cells that identified expected core essential genes

(<https://www.biorxiv.org/content/early/2018/05/16/323436>). In addition, we have also performed our own screens in human ES cells with similar results to Ihry et al. Furthermore, other groups (Giacomelli et al, Nature Genetics 2018, PMID 3022464; <https://www.biorxiv.org/content/early/2018/09/03/407767>) reported that TP53 enrichment can be observed in non-ES cell lines, as well as in RNAi screens presumably because of shRNA integration. We have now carefully reassessed the wording of the relevant sections and included further details and references to better justify our statements.

Of minor note: HeLa cells are described in the manuscript as "mutant", though they have WT p53 (albeit nonfunctional due to other reasons). This should be corrected.

We have now corrected this in the text.

2nd Editorial Decision

1st February 2019

Thank you again for sending us your revised manuscript. We have now heard back from the three reviewers who were asked to evaluate your revised study. As you will see below the reviewers are satisfied with the modifications made and think that the study is now suitable for publication. I am therefore pleased to inform you that your paper has been accepted for publication.

REFeree REPORTS

Reviewer #1:

The authors addressed the points in my original review.

Reviewer #2:

The authors have addressed all points raised by the reviewers and have thus strengthened their manuscript. I have no further comments to raise and as such as I happy for this manuscript to proceed with a publication.

Reviewer #3:

In the view of this reviewer, the authors have addressed all the raised concerns in a convincing manner. I thus recommend manuscript for publication.

Corresponding Author Name: Jason Moffat
Journal Submitted to: Molecular Systems Biology
Manuscript Number: MSB-18-8679